# Curcumin Ameliorates Benzo[a]pyrene-Induced DNA Damages in Stomach Tissues of Sprague-Dawley Rats

**DOI:** 10.3390/ijms20225533

**Published:** 2019-11-06

**Authors:** Kyeong Seok Kim, Na Yoon Kim, Ji Yeon Son, Jae Hyeon Park, Su Hyun Lee, Hae Ri Kim, Boomin Kim, Yoon Gyoon Kim, Hye Gwang Jeong, Byung Mu Lee, Hyung Sik Kim

**Affiliations:** 1School of Pharmacy, Sungkyunkwan University, Gyeonggi-do, 2066, Seobu-ro, Suwon 16419, Korea; caion123@nate.com (K.S.K.); twiase@naver.com (J.Y.S.); sky3640@naver.com (J.H.P.); lydia7334@skku.edu (S.H.L.); kimhaeri56@daum.net (H.R.K.); boomin@hanmail.net (B.K.); bmlee@skku.edu (B.M.L.); 2College of Pharmacy, Dankook University, Chungnam, 119, Cheonan 31116, Koreakyg90@dankook.ac.kr (Y.G.K.); 3College of Pharmacy, Chungnam National University, 99, Daehak-ro, Yuseong-gu, Daejeon 34134, Korea; hgjeong@cnu.ac.kr

**Keywords:** curcumin, benzo(a)pyrene, gastrointestinal cancer, BPDE-I-DNA adduct, 8-hydroxydeoxy guanosine

## Abstract

Benzo[a]pyrene (BaP) is a well-known carcinogen formed during the cooking process. Although BaP exposure has been implicated as one of the risk factors for lung cancer in animals and humans, there are only limited data on BaP-induced gastrointestinal cancer. Therefore, this study investigated the protective effects of curcumin on BaP-induced DNA damage in rat stomach tissues. BaP (20 mg/kg/day) and curcumin (50, 100, or 200 mg/kg) were administered daily to Sprague-Dawley rats by oral gavage over 30 days. Curcumin was pre-administered before BaP exposure. All rats were euthanized, and liver, kidney, and stomach tissues were removed at 24 h after the last treatment. We observed that aspartate aminotransferase (AST), alanine aminotransferase (ALT), and glucose levels were significantly reduced in rats treated with high dose co-administration of curcumin (200 mg/kg) compared to BaP alone. The expression levels of cytochrome P450 (CYP) 1A1 and CYP1B1 were significantly increased in the liver of rats treated with BaP. However, co-administration of curcumin (200 mg/kg) with BaP markedly reduced CYP1A1 expression in a dose-dependent manner. Furthermore, plasma levels of BaP-diolepoxide (BPDE) and BaP metabolites were significantly reduced by co-administration of curcumin (200 mg/kg). Additionally, co-administration of curcumin (200 mg/kg) with BaP significantly reduced the formation of BPDE-I-DNA and 8-hydroxydeoxy guanosine (8-OHdG) adducts in the liver, kidney, and stomach tissues. The inhibition of these adduct formations were more prominent in the stomach tissues than in the liver. Overall, our observations suggest that curcumin might inhibit BaP-induced gastrointestinal tumorigenesis and shows promise as a chemopreventive agent.

## 1. Introduction

Benzo[a]pyrene (BaP) is considered a major pollutant in the environment. BaP is a polyaromatic hydrocarbon (PAH) found in coal tar, coal-processing waste products, petroleum sludge, asphalt, and tobacco smoke [1,2]. Exposure to BaP causes various adverse health effects including cancer development, immunosuppression, teratogenicity, and hormonal dysfunctions [3,4]. With the exception of occupational exposure, the major route of BaP exposure to human is via contaminated foods because BaP can be produced by the pyrolysis of amino acids, fatty acids, and carbohydrates during the cooking process [5,6]. Therefore, the probability of BaP exposure to humans through the consumption of grilled meats, water, and smoked fishes is very high [7,8]. In addition to specific components of the diet, the cooking process is also associated with increased risk of gastrointestinal cancer, which is causatively related to the formation of BaP-DNA adducts in the stomach [9,10,11]. Furthermore, dietary habits have been linked to high incidence of gastric cancer in many areas of the world [12,13]. Previous studies indicated that BaP has been proven to cause cancer in the skin, lung, mammary glands, and forestomach tissues of experimental animals [14,15,16]. Although the molecular mechanism of BaP-induced gastric cancer has been elucidated, many studies have only indirectly evaluated the correlation between BaP exposure and gastric cancer. Therefore, the association remains unclear.

Gastric cancer is the fourth most common type of cancer and the second leading cause of cancer death in the world [17]. It is a major cause of cancer mortality in Korea, Japan, and India. The development of gastric cancer is a complex, multistep process involving multiple genetic and epigenetic alterations in oncogenes, tumor suppressor genes, DNA repair genes, cell cycle regulators, and signaling molecules [18,19]. At the early stage of gastric cancer, atrophy and intestinal metaplasia might be involved in the development of gastric adenocarcinoma [20]. The fact that diet also plays an important role in the etiology of gastric cancer offers a scope for nutritional chemoprevention. The development of multitargeted preventive and therapeutic strategies for gastric cancer is a major challenge for the future. Fruits, vegetables, common beverages, and several medicinal herbs with diversified pharmacological properties have been shown to be a rich source of cancer chemopreventive agents [21].

Curcumin (diferuloylmethane), a polyphenol compound, is an active ingredient of turmeric (*Curcuma longa*). Curcumin is a polyphenol of interest for application as a chemopreventive agent. The chemopreventive potential of curcumin in preventing the process of carcinogenesis was previously demonstrated by various studies [22,23,24]. Curcumin shows beneficial effects in many cancers including forestomach, colorectal cancer, breast cancer, skin cancer, and oral cancer [25,26,27,28]. Furthermore, curcumin has been known to retard the formation of DNA adducts by carcinogen exposure, and thus delay the process of tumorigenesis in several animal models [29,30]. However, data on the effect of curcumin on DNA adduct formation through the in vivo study of gastric cancer are limited. Therefore, the aim of the present study was to determine the protective effect of curcumin on BaP-induced gastric carcinogenesis.

## 2. Results

### 2.1. Changes in Body Weight, Organ Weights, and Blood Biochemistry

Over the experimental period, all animals were weighed periodically. Animals treated with either BaP or BaP in combination with curcumin exhibited a slight, but significant reduction in body weight compared to controls. However, there was no difference among the treated groups (Figure 1).

Liver weight was significantly reduced by the co-administration of BaP with curcumin (Table 1). Serum alanine aminotransferase (ALT) and aspartate aminotransferase (AST) activities, as well as blood urea nitrogen (BUN) and glucose levels, were significantly increased in the BaP-treated group compared with control group. In contrast, co-administration of curcumin (100 or 200 mg/kg) with BaP exhibited significant reduction in AST, ALT, BUN, and glucose (Table 2). These results were highly correlated with the liver and kidney histopathological changes, whereby oral administration of BaP slightly increased hepatic and renal damage in rats. As shown in Figure 2, hepatic tissues displayed cell infiltration, mononuclear cells, and multifocal cells in rats. The induction in non-carcinogenic target tissues kidney showed only focal nephropathy (Table 3). We did not detect histopathological changes in the stomach.

### 2.2. Effect of Curcumin on Formation of BaP and its Metabolites in Serum of Rats

To detect BaP and its metabolites, a spectrum of different wavelengths was tested to detect BaP or BaP metabolites. The eluted BaP, as well as metabolite peak, was confirmed based on the internal standard (IS). For highest sensitivity of detection, BaP and its metabolites were separated by liquid chromatography tandem mass spectrometry (LC/MS/MS). Chromatographic profiles were almost baseline with the elution of BaP and metabolites, probably due to retaining the BaP and its metabolites on the resin under such conditions (Figure 3). We could detect the BaP metabolites in the serum including 3-hydroxy BaP (3-OH-BaP), 7-hydroxy BaP (7-OH-BaP), and BPDE. The co-administration of curcumin significantly reduced the concentration of 7-OH-BaP and BPDE when compared to the BaP alone group. In contrast, serum concentration of parent compound BaP was increased by co-administration of curcumin in a dose-dependent manner (Figure 4).

### 2.3. Effect of Curcumin on Expression of Hepatic CYPs in Rats

The induction of cytochrome P450 (CYP)1A1 and CYP1B1 by environmental xenobiotic chemicals or endogenous ligands through the activation of the aryl hydrocarbon receptor (AhR) has been implicated in a variety of cellular processes related to cancer, such as transformation and tumorigenesis [31]. In our study, we measured the expression of CYP1A1 and CYP1B1 in the liver and stomach of experimental rats. In the BaP alone group, hepatic expression of CYP1A1 and CYP1B1 levels were highly increased, especially CYP1A1 levels (up to eight-fold compared with control). However, the co-administration of curcumin (200 mg/kg) with BaP markedly reduced the expression of CYP1A1 and CYP1B1 in the liver of rats (Figure 5). In the stomach tissues, CYP1A1 and CYP1B1 expression levels following the co-administration of curcumin were very similar to levels in the liver.

### 2.4. Effect of Curcumin on BaP-Induced DNA Damage in Rats

To determine whether curcumin suppressed the BaP-induced DNA adduct formation, BPDE-I-DNA adduct levels were quantitated using ELISA. As shown in Figure 6, the BPDE-DNA adduct levels were significantly increased in the liver, kidney, and stomach of rats following BaP exposure, whereas the co-administration of curcumin significantly reduced BPDE-I-DNA adduct formation in a dose-dependent manner. Especially, BPDE-DNA adduct levels were inhibited in the stomach tissues following the co-administration of curcumin (200 mg/kg). These data are similar to the expression level of hepatic phase I enzyme levels because a significant inhibition of BaP-induced CYP1A1 and CYP1B1 was observed in the liver of curcumin-treated groups. Since curcumin markedly inhibited CYP1A1 and CYP1B1 from forming BPDE or other major BaP metabolites, it could also alter the DNA oxidative damage in the target organs. Therefore, levels of the oxidative DNA damage marker, 8-OHdG, were measured in response to reactive metabolites generated during BaP metabolism. The BaP-treated group showed a significant elevation of 8-OHdG levels in the liver and stomach tissues. However, significant inhibition of 8-OHdG levels was found in curcumin co-administered groups compared with BaP alone (Figure 7). Consequently, the sum total of the formation of DNA damage was reduced in a dose-dependent manner by the co-administration of curcumin when compared to BaP alone in rats.

## 3. Discussion

The present study investigated the protective role of curcumin against BaP-induced DNA damage in the target organs of rats. Although active metabolites BaP cause DNA damage in various tissues, little is known about the potential role of BaP-induced DNA damage in stomach carcinogenesis. Hence, we investigated whether curcumin suppressed BaP-induced DNA adduct formation in the target organs. Taken together, this study provides new insights into the novel mechanisms of curcumin acting on the regulation of BaP metabolism, which may be important for understanding BaP-induced carcinogenesis. Our data indicated that curcumin administration significantly inhibited BaP-induced DNA damage in the stomach tissues compared with other organs by inhibiting BaP metabolism. Therefore, we suggest that dietary curcumin may inhibit BaP-induced gastric tumorigenesis by reducing the formation of active BaP metabolites, as well as increasing the detoxification of BaP metabolites.

The molecular mechanism of BaP-induced carcinogenesis is closely related to DNA adduct formation and oxidative DNA damage through metabolic activation first oxidized by CYP1A1 and CYP1B1 to form intermediate metabolites, BaP-7,8-epoxide [32,33]. Through increased metabolic activation, BaP-7,8-epoxide changes to its ultimate carcinogen, anti-BPDE-I, which covalently binds to DNA to form BPDE-DNA adducts [34]. DNA adducts induced by BaP in various organs have been studied as biomarkers for PAH exposure or toxicity [35]. Although DNA adducts are used as biomarkers for chemical exposure to clarify genotoxic and nongenotoxic carcinogens, they may not accurately predict with certainty their carcinogenic potential in the target organs. A previous study showed that acute exposure to BaP by oral gavage leads to similar levels of DNA adduct formation in the lungs and livers, which are BaP-bioactivating organs [36]. The present study was undertaken to further explore the gastric carcinogenesis underlying such distinct tissue-specific responses using a subchronic BaP exposure model.

BaP also produces a few quinone derivatives and reactive oxygen species (ROS), which are mutagens and carcinogens, during BaP metabolic activation [37]. ROS was generated as intermediate metabolites and damages proteins, lipids, and DNA [32,38,39]. In particular, the formation of 8-OHdG has been widely used as a biomarker of oxidative DNA damage [40]. We found that the co-administration of curcumin resulted in increased plasma concentration of unmetabolized BaP and decreased concentration of intermediate metabolites including 7-OH-BaP and BPDE, indicating that curcumin inhibited the active metabolites formation from BaP. Therefore, our data support that the formation of DNA-adduct concentration may have been reduced in the blood. As expected, BaP-induced DNA adducts patterns were concurrent with protein expression levels of CYP1A1 and CYP1B1 and its active metabolites formation. Although there is a linear correlation between mutation frequency and tumor incidence in target organs after exposure of animals to BaP, the levels of BaP-DNA adduct formation did not distinguish target (lung, spleen, and forestomach) from non-target organs (liver, colon and glandular stomach) in mice following oral administration of BaP [41]. This study was very similar to our present data that oral exposure to BaP significantly increased BPDE-I-DNA adduct formation in the stomach and liver tissues. Induction of CYP1A1 and CYP1B1 by environmental xenobiotic chemicals or endogenous ligands through the activation of the aryl hydrocarbon receptor (AhR) has been implicated in a variety of cellular processes related to cancer, such as transformation and tumorigenesis [31,41]. Here, we investigated the effects of curcumin on expression of CYP1A1 and CYP1B1 in the liver and stomach of rats treated with BaP for 30 days. We found that curcumin significantly inhibited CYP1A1 expression in the liver and stomach to protect the formation of BaP active metabolites.

The underlying mechanisms of chemoprevention may be induced not only because curcumin possesses good ROS scavenging abilities, but curcumin may also be capable of inhibiting BaP metabolism enzymes. Sehgal et al. (2013) demonstrated that curcumin suppresses BaP-induced DNA damage in liver and lung of mice by increasing the activity of ethoxyresorufin-O-deethylase (EROD) [42]. Zhu et al. (2014) found that curcumin inhibited CYP1A1 and CYP1A1-catalyzed 7,8-diol-BaP-epoxidation, which is the terminal reaction leading to the ultimate carcinogenic product, diolepoxide, in lung epithelial cells [43]. The data from the above studies suggest that the protective effects of curcumin against BaP-induced DNA damage and precancerous changes in cells are initiated upstream from CYP1A1. Therefore, the protective effects of curcumin against BPDE-DNA adduct formation in the stomach is related to inhibition of CYP1A1 and CYP1A2 expression, rather than inhibition of the enzyme activity itself. These results suggest that while unrepaired DNA adducts were formed in the stomach, other factors, such as alterations in critical molecular processes implicated in cancer formation, may play a role in the selective targeting of the stomach tissue following BaP exposure.

Another mechanism of inhibition of DNA adduct formation by curcumin is the inactivation of BaP-derived reactive metabolites such as ROS and BaP-quinone derivatives. In the present study, we showed a clear inverse relationship between curcumin dose and the level of BPDE-DNA adducts. However, curcumin is indeed capable of protecting against ROS formation. For the DNA damages and the precancerous pathologic changes in cells that can lead to carcinogenesis, this is also the potential mechanism for the chemopreventive effects of curcumin during carcinogenesis induced by BaP and other procarcinogens. Therefore, curcumin protects the BaP-induced 8-OHdG formation in target tissues including the liver, kidney, and stomach.

## 4. Materials and Methods

### 4.1. Chemicals and Reagents

BaP, nuclease P1, and curcumin were purchased from Sigma-Aldrich (St. Louis, MO, USA). Alkaline phosphatase was purchased from Takara Bio Inc. (Shiga, Japan). All other western blot reagents were from Merck Millipore (Burlington, MA, USA). Primary antibodies (CYP 1A1, CYP 1B1, and β-actin) and a horseradish peroxidase (HRP)-conjugated secondary antibody were purchased from Santa Cruz Biotechnology (Santa Cruz, CA, USA). The Genomic DNA isolation kit was purchased from Qiagen (Venlo, Netherlands). All enzyme-linked immunosorbent assay (ELISA) kits were purchased from Cell Biolabs Inc. (BPDE-DNA adduct kit no. STA-357, 8-OHdG kit no. STA-320; San Diego, CA, USA). Hematoxylin and eosin were purchased from Dako (Glostrup, Denmark).

### 4.2. Animal Experiments

Sprague-Dawley rats (five-week-old males, weighing 140–150 g) were obtained from Orient-Bio (Seongnam-si, Korea). All animals were maintained in a specific pathogen free (SPF)-conditioned room with a 12-h light/dark cycle. The ambient air temperature and relative humidity was set to 23 ± 2 °C and 55%, respectively. Before the experiments, all animals were checked for any overt signs of illness and only healthy animals were selected. Rodent chow and water were supplied *ad libitum*. The experimental protocol was approved by Sungkyunkwan University Laboratory Animal Care Service (SKKU-2013-000105, March 23, 2013) in accordance with the Ministry of Food and Drug Safety (MFDS) Animal Protection of Korea (Oh-Song, Korea). Animals in each group (*n* = 6) were randomly divided into five groups: (1) Control group, where rats were treated with corn oil for 30 days; (2) BaP-treated group, where rats were administered orally BaP (20 mg/kg) dissolved in corn oil for 30 days; and (3) co-treatment of BaP and curcumin groups, where curcumin (50, 100, or 200 mg/kg) was administered orally before BaP (20 mg/kg) exposure for 30 days. The inclusion of BaP concentrations was chosen based on other studies [44,45]. The BaP doses employed in our study are relevant to human exposure scenarios, as the levels of pulmonary arterial hypertension (PAH) are similar to levels that would be acquired in the diet [34]. If a young adult weighing 50 kg were to eat a half-pound (0.23 kg) of fried chicken (containing 5 μg/kg) or charcoal-broiled steak (containing 9 μg/kg) every day, it translates to a PAH dose of 30–50 ng/kg/day [46]. Some studies have reported a total PAH intake of 14 μg/kg/day [47] and 59.2 μg/ kg/day [48], and a more recent report listed an intake of 371 μg/person/day [1] The treatments were administered daily at the same time, and animals were acclimated for 1 h before treatment.

### 4.3. Biochemical Parameters in Serum

Blood was collected from the abdominal aorta and collected in 15-mL plain tubes. Within 1 h of collection, blood samples were then centrifuged at 3000× *g* for 10 min to collect serum. The sera were immediately stored at −80 °C to analyze blood urea nitrogen (BUN) and glucose levels, as well as ALT and AST activities, using an Olympus AU400 chemistry analyzer (Tokyo, Japan).

### 4.4. Determination of the BaP and BaP Metabolites in Plasma

#### 4.4.1. LC/MS/MS Analysis

For the analysis of BaP, 3-hydroxy BaP (3-OH BaP), 7-hydroxy BaP (7-OH BaP), and BaP-7,8-dihydrodiol-9,10-epoxide (BPDE) concentration in plasma, a LC/MS/MS method was developed. A liquid chromatographic system (Dionex Ultimate^®^ 3000, Thermo Fisher Scientific Inc., Boston, MA, USA) equipped with a pump, autosampler, and column compartment was used and connected to a quadrupole tandem mass spectrometer (AB SCIEX API 3200, Applied Biosystems Sciex, Toronto, Ontario, Canada) equipped with an atmospheric pressure chemical ionization (APCI) source. System control and data analyses were carried out with Analyst 1.5.2. Chromatographic separation was performed using a Unison UK C8 (75 × 2.0 mm, 3 μm, Imtakt, Kyoto, Japan) protected by a guard column (4.0 × 2.0 mm, Phenomenex, Torrance, CA, USA). The column oven temperature was 40 °C. The mobile phase of 0.1% formic acid in water for the A pump and 0.1% formic acid in acetonitrile for the B pump (2:98, *v*/*v*) was run at a flow rate of 400 mL/min. The injection volume was 10 μL and the total analysis run time was 2 min. The atmospheric pressure chemical ionization (APCI) mass spectrometer was operated in the positive ion mode. Multiple reaction monitoring (MRM) of the precursor-product ion transitions from m/z 253.0 to m/z 250.0 for BaP, from m/z 269.0 to m/z 252.0 for 3-OH BaP, from m/z 269.0 to m/z 241.0 for 7-OH BaP, and from m/z 304.1 to m/z 258.2 for BPDE were used for quantitation. Collision energy of BaP, 3-OH BaP, 7-OH BaP, and BPDE was 77, 40, 30, and 17 V, respectively. The optimized conditions were as follows: Nebulizer current (5 μA), curtain gas (30 psi), collision gas (5 psi), ion spray voltage (5500 V), source temperature (300 °C), and ion source gas 1 (30 psi).

#### 4.4.2. Preparation of Stock Solution and Standard Solution

A stock solution of BaP, 3-OH BaP, 7-OH BaP, and BPDE was prepared in acetonitrile at 1.0 mg/mL and dilution solutions of stock solution were used with 0.1% formic acid in 50% acetonitrile. Standard solutions of BaP, 3-OH BaP, 7-OH BaP, and BPDE were prepared by spiking with an appropriate volume of the diluted stock solution, giving final concentrations of 10, 50, 100, 500, and 1000 ng/mL. The internal standard (IS, cryptotanshinone) solution was prepared in acetonitrile and diluted with 0.1% formic acid in 50% acetonitrile to give a final concentration of 1 μg/mL.

#### 4.4.3. Sample Preparation

Plasma was prepared through a liquid-liquid extraction method. A 50-μL aliquot of plasma was spiked with 5 μL of IS (1 μg/mL) in a 1.5-mL polyethylene micro tube. Next, 500 µL of ethyl acetate was added as an extraction solvent to the tubes. The tubes were vortex-mixed for 1 min and centrifuged (Hanil Science Industrial Co. Ltd., Incheon, South Korea) for 5 min at 12,000× *g*. The organic layer was transferred to another microtube and evaporated under nitrogen gas at 40 °C in a MG 2100 Eyela dry thermo bath (Rikakikai Company, Tokyo, Japan). Then, 100 µL of the 0.1% formic acid in 98% acetonitrile was added as mobile phase to reconstitute the residue and 100 µL of the supernatant was directly injected to the HPLC system. The injection volume was kept at 10 µL.

#### 4.4.4. Calibration Curve

The calibration standards used were 10, 50, 100, 500, 1000, and 2000 ng/mL of BaP, 3-OH BaP, 7-OH BaP, and BPDE in plasma. These plasma samples were extracted as described above in the sample preparation section. Calibration curves of BaP, 3-OH BaP, 7-OH BaP, and BPDE were constructed using the weighted regression method (R = 0.999, 1/x^2^, weighting) by plotting the peak area ratio of analytes and IS versus the concentrations. The lower limit of quantitation (LLOQ) was 10 ng/mL for all four compounds.

### 4.5. Protein Preparation and Western Blot Analysis

The liver and stomach were detached immediately, rinsed with cold phosphate-buffered saline (PBS), and stored at −80 °C. Tissues were thawed and homogenized using PRO-PREP^TM^ protein extract solution (iNtRON, Seongnam, Korea). Protein was quantitated using the protein assay reagent (Pierce, Rockford, IL, USA) according to the manufacturer’s instructions. Extracted proteins were denatured by boiling at 95 °C for 5 min in sample buffer (0.5 M Tris-HCL, pH 6.8, 4% SDS, 20% glycerol, 0.1% bromophenol blue, 10% β-mercaptoethanol). Protein samples (30 µg) were run on 8~12% SDS-polyacrylamide gel electrophoresis (SDS-PAGE) at 100 V for 90 min using running buffer (25 mM Tris, 192 mM glycine, 0.1% SDS). The proteins were electrophoretically transferred to polyvinylidene difluoride (PVDF) membranes (Millipore, Burlington, MA, USA) at 100 V for 90 min in transfer buffer (25 mM Tris-HCl pH 9.5, 192 nM glycine, 20% Methanol). The membranes were then blocked using blocking buffer (TNT buffer containing 5% skim milk) for 1 h. Next, membranes were incubated overnight at 4 °C with primary antibodies (CYP1A1, CYP1B1, and β-actin). After washing for 60 min with TNT buffer (10 mM Tris-HCl, pH 7.6, 100 mM NaCl, and 0.5% Tween 20), the membranes were incubated for 60 min with HRP-conjugated goat anti-rabbit IgG (1:10,000) or anti-mouse IgG (1:8000), and then washed for 90 min with TNT buffer. The blots were developed using an Enhanced chemiluminescence (ECL)-plus kit (Amersham Biosciences, Amersham, Buckinghamshire, UK).

### 4.6. Quantitation of BPDE-DNA Adducts

The ELISA kit was purchased from Cell Biolabs Inc. (San Diego, CA, USA). The formation of BPDE-DNA adducts was determined using enzyme-linked immunoassay following the manufacturer’s protocol. DNA was extracted from the stomach and liver using QIAamp DNA Mini Kit. Briefly, 100 µL of unknown DNA sample and standards were incubated for 2 h at 37 °C. The solutions were removed and washed two times using PBS, and 100 µL of anti-BPDE-I antibody was added and incubated for 1 h at room temperature. Then, wells were washed five times with washing buffer and each well was blocked using blocking solution. After removal of solutions and washing with washing buffer three times, wells were incubated with secondary antibody at room temperature for 1 h and then washed three times. Next, 100 µL of 3,3′,5,5′-tetramethylbenzidine (TMB) solution was added to each well and incubated for 20 min at room temperature. The reactions were stopped using stop solution, and absorbance was measured using a microplate reader (Molecular Devices, model VERSA max™, San Jose, CA, USA) at 450 nm.

### 4.7. Determination of 8-hydroxydeoxyguanosine

The ELISA kits for quantitating 8-hydroxydeoxyguanosine (8-OHdG) were purchased from Cell Biolabs Inc. The assay was performed following instructions of the manual. Briefly, DNA was isolated from tissues and then incubated at 95 °C for 5 min and rapidly chilled on ice. DNA samples were digested to nucleosides by incubating the denatured DNA with 10 units of nuclease P1 for 2 h at 37 °C in 20 mM sodium acetate (pH 5.2), treated with 10 units of alkaline phosphatase (Takara Bio Inc.) for 15 min at 37 °C, and then incubated at 50 °C for 15 min in 100 mM Tris buffer (pH 7.5). The reaction mixture was centrifuged for 5 min at 6000× *g* and the supernatant were used for the assay. Simply, 50 µL of unknown samples or standards were incubated at room temperature for 10 min, then 50 µL of the diluted anti-8-OHdG antibody was added and incubated at room temperature for 1 h on an orbital shaker. After washing three times, 100 µL of diluted secondary antibody-enzyme conjugate was added and incubated at room temperature for 1 h before washing three times. Substrate solution (100 µL) was added to each well and incubated at room temperature for 20 min. The enzyme reaction was stopped by adding 100 µL of stop solution, and absorbance was measured using the microplate reader at 450 nm.

### 4.8. Statistical Analysis

All values are expressed as the mean ± standard error of the mean (SEM). Statistical analysis was performed using *SigmaStat* (SPSS Inc., Chicago, IL, USA). Statistically significant differences were determined using one-way analysis of variance (ANOVA) followed by Dunn’s test. A *p*-value < 0.05 was considered statistically significant.

## 5. Conclusions

In summary, our results showed that curcumin has a protective effect against BaP-induced DNA damages by the inhibiting expression of phase I enzymes in the liver and stomach, ultimately inhibiting DNA adducts formation in the target organs. Especially, stomach tissue was perfectly protected DNA damages by curcumin during BaP-induced carcinogenesis. Our results suggest that curcumin significantly protected against gastric carcinogenesis through dietary exposure of BaP, which is related to the inhibition of BaP metabolic activation and ROS generation.

## Figures and Tables

**Figure 1 ijms-20-05533-f001:**
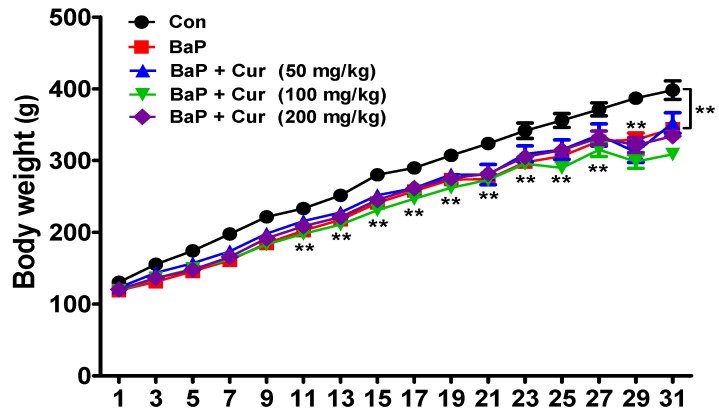
Effects of curcumin on body weight changes in rats treated with benzo(a)pyrene. Rats were orally administrated benzo[a]pyrene (BaP) alone (20 mg/kg) or in combination with curcumin (50, 100, or 200 mg/kg) for 30 days. The control group was administered vehicle only. Values are indicated by mean ± SEM of six animals. Significant difference from the control group at ** *p* < 0.01. Cur: curcumin.

**Figure 2 ijms-20-05533-f002:**
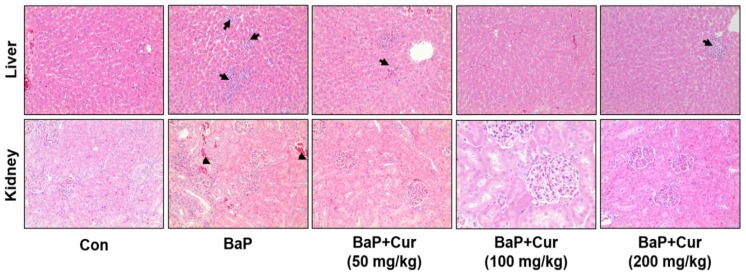
Effects of curcumin on histopathological changes in liver and kidney of rats treated with benzo[a]pyrene (BaP). Rats were orally administrated (BaP alone (20 mg/kg) or in combination with curcumin (50, 100, or 200 mg/kg) for 30 days. The control group was administered vehicle only. Representative histology of hematoxylin and eosin (H&E)-stained liver and kidney sections from experimental groups. Arrows displayed cell infiltration, mononuclear cells, and multifocal cells. Arrowheads indicated focal nephropathy. Cur; curcumin. Original magnification: ×100.

**Figure 3 ijms-20-05533-f003:**
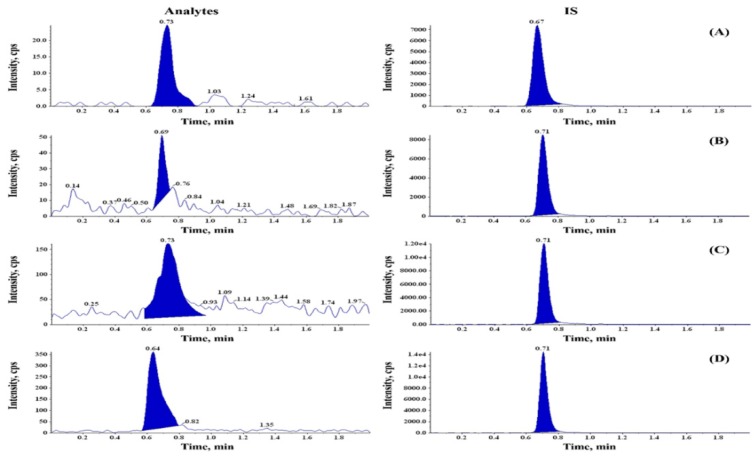
Separation of the benzo[a]pyrene and its metabolites in serum of rats. The analyses of the benzo[a]pyrene (BaP) metabolites were detected using liquid chromatography tandem mass spectrometry (LC/MS/MS). Mobile phase; D.W (0.1% formic acid): Acetonitrile (0.1% formic acid) = 2:98, calibration curve: R = 0.99, with gradient program (2:98, D.W, containing 0.1% formic acid: Acetonitrile containing 1% formic acid) as a mobile phase. Chromatographic profiles as almost baseline without the elution of BaP and metabolites, probably due to retaining the BaP and its metabolites on the resin under such conditions. Left panels indicated the typical LC/MS/MS chromatogram of (**A**) BaP; (**B**) 3-OH BaP; (**C**) 7-OH BaP; (**D**) BaP-diolepoxide (BPDE). Right panels indicated the typical LC/MS/MS chromatograms of internal standard (cryptotanshinone).

**Figure 4 ijms-20-05533-f004:**
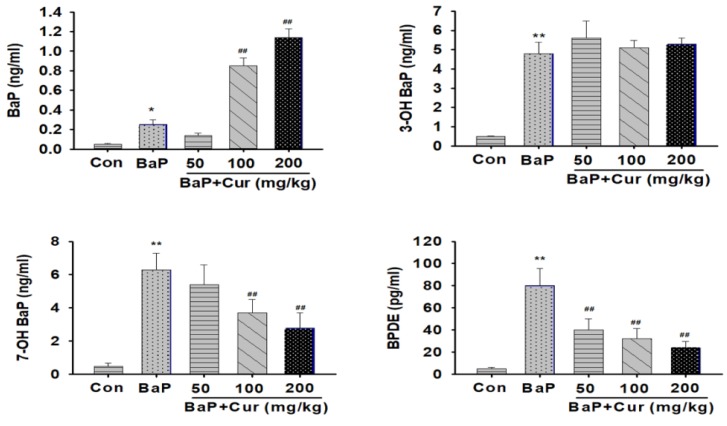
Effect of curcumin on the formation of major benzo[a]pyrene metabolites in serum of rats treated orally with benzo(a)pyrene. Rats were orally administrated benzo[a]pyrene (BaP) alone (20 mg/kg) or in combination with curcumin (50, 100, or 200 mg/kg) for 30 days. The analysis of BaP metabolite was detected using LC/MS/MS. The control group was administered vehicle only. Values are indicated by mean ± SD of six animals. Significant difference from the control group at * *p* < 0.05; Significant difference from the control group at ** *p* < 0.01; Significant difference from the BaP alone group at ^##^
*p* < 0.01. Cur: curcumin.

**Figure 5 ijms-20-05533-f005:**
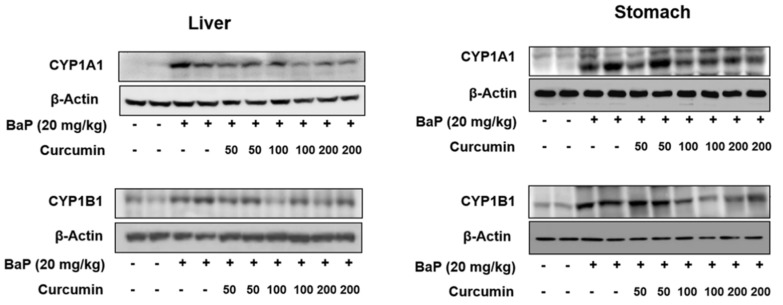
Effect of curcumin on the expression of cytochrome P450 (CYP)1A1 and CYP1B1 in liver and stomach of rats treated with benzo[a]pyrene (BaP). Rats were orally administrated BaP alone (20 mg/kg) or in combination with curcumin (50, 100, or 200 mg/kg) for 30 days. The control group was administered vehicle only. Representative bands of western blot for CYP1A1 and CYP1B1 were shown. β-Actin was used as endogenous control to normalize the data.

**Figure 6 ijms-20-05533-f006:**
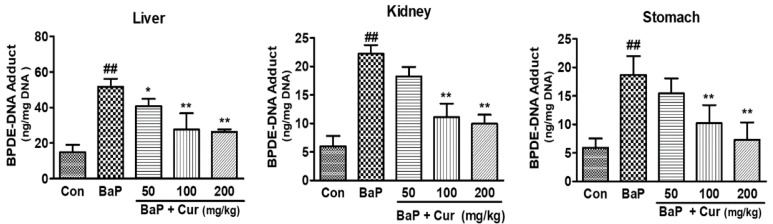
Effect of curcumin on the BPDE-DNA adducts formation in the target organs of Sprague-Dawley rats treated with benzo[a]pyrene (BaP). Rats were orally administrated BaP alone (20 mg/kg) or in combination with curcumin (50, 100, or 200 mg/kg) for 30 days. The control group was administered vehicle only. Values are indicated by mean ± SD of six animals. Significant difference from the control group at ^##^
*p* < 0.01; Significant difference from the BaP alone group at *****
*p* < 0.05; Significant difference from the BaP alone group at ** *p* < 0.01. Cur: curcumin.

**Figure 7 ijms-20-05533-f007:**
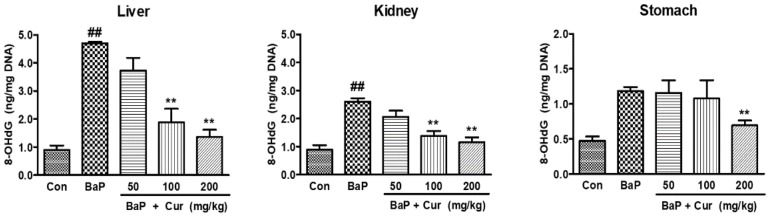
Effect of curcumin on the 8-hydroxydeoxyguanosine (8-OHdG) formation in the target organs of Sprague-Dawley rats treated with benzo[a]pyrene (BaP). Rats were orally administrated BaP alone (20 mg/kg) or in combination with curcumin (50, 100, or 200 mg/kg) for 30 days. The control group was administered vehicle only. Values are indicated with mean ± SD of six animals. Significant difference from the control group at ^##^
*p* < 0.01; Significant difference from the BaP alone group at ** *p* < 0.01. Cur: curcumin.

**Table 1 ijms-20-05533-t001:** Effects of Curcumin on Organ Weight Changes in Sprague-Dawley Rats.

Groups	Organ Weights (g)				
	Liver	Kidney	Stomach	Spleen	Testis
Control	3.6 ± 0.11	0.35 ± 0.01	0.52 ± 0.02	0.19 ± 0.01	0.39 ± 0.01
BaP (20 mg/kg)	3.8 ± 0.10	0.35 ± 0.01	0.53 ± 0.03	0.23 ± 0.01	0.43 ± 0.02
BaP + Cur. (50 mg/kg)	3.2 ± 0.07 *	0.34± 0.01	0.48 ± 0.03	0.20 ± 0.01	0.39 ± 0.01
BaP + Cur. (100 mg/kg)	3.1 ± 0.09 *	0.33 ± 0.02	0.54 ± 0.04	0.21 ± 0.01	0.42 ± 0.01
BaP + Cur. (200 mg/kg)	2.9 ± 0.09 *	0.36 ± 0.01	0.55 ± 0.02	0.21 ± 0.02	0.42 ± 0.02

Rats were orally administrated corn oil (control) or benzo[a]pyrene (BaP) alone (20 mg/kg) or in combination with curcumin (50, 100, or 200 mg/kg) for 30 days. The control group was administered the vehicle only. Values are indicated with mean ± SD of six animals. Significant difference from the BaP alone group at * *p* < 0.05.

**Table 2 ijms-20-05533-t002:** Effects of Curcumin on Serum Biochemical Parameters in Sprague-Dawley Rats.

Groups	Biochemical Parameters			
	ALT (U/L)	AST (U/L)	BUN (mg/dL)	Glucose (mg/dL)
Control	53.0 ± 3.3	120.2 ± 8.0	11.9 ± 0.3	134.67 ± 7.4
BaP (20 mg/kg)	61.4 ± 2.1 *	144.4 ± 8.4 *	20.7 ± 1.8 *	184.65 ± 6.8 *
BaP + Cur (50 mg/kg)	54.3 ± 6.0	127.6 ± 8.5	15.4 ± 0.7	144.1 ± 10.1
BaP + Cur (100 mg/kg)	45.9 ± 5.8 ^#^	121.2 ± 5.9 ^#^	12.9 ± 1.1 ^##^	115.0 ± 14.9 ^##^
BaP + Cur (200 mg/kg)	36.7 ± 1.9 ^##^	122.0 ± 6.2 ^#^	12.0 ± 0.3 ^##^	104.8 ± 8.3 ^##^

Rats were orally administrated corn oil (control) or benzo[a]pyrene (BaP) alone (20 mg/kg) or in combination with curcumin (50, 100, or 200 mg/kg) for 30 days. The control group was administered the vehicle only. Values are indicated with mean ± SD of six animals. ALT, alanine aminotransferase; AST, aspartate aminotransferase; BUN, blood urea nitrogen. Significant difference from the control group at * *p* < 0.05; Significant difference from the BaP alone group at ^#^
*p* < 0.05; Significant difference from the BaP alone group at ^##^
*p* < 0.01.

**Table 3 ijms-20-05533-t003:** Histopathological Changes in Liver and Kidney of Sprague-Dawley Rats.

Groups	Control	BaP	BaP + Cur (50 mg/kg)	BaP + Cur (100 mg/kg)	BaP + Cur (200 mg/kg)
No. of animals examined	6	6	6	6	6
**Liver**					
No. of specific lesions	5	0	2	2	2
Cell infiltration, mononuclear cells, multifocal cells	1	6	4	4	4
Minimum	1	6	4	3	4
Mild	0	0	0	1	0
**Kidney**					
No. of specific lesions	4	1	4	4	6
Focal nephropathy	2	5	2	1	0
Minimum	0	5	2	1	0
Cell infiltration, lymphocytic	0	0	0	0	0
Minimum	0	0	0	0	0

Rats were orally administrated corn oil (control) or benzo[a]pyrene (BaP) alone (20 mg/kg) or in combination with curcumin (50, 100, or 200 mg/kg) for 30 days. The control group was administered the vehicle only. Values are indicated with mean ± SD of six animals. Minimum: <10 foci; mild: 10–15 foci in a specimen. Cur; curcumin.

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
