# Peer review of "Curcumin Ameliorates Benzo[a]pyrene-Induced DNA Damages in Stomach Tissues of Sprague-Dawley Rats"

_ijms, 2019, doi:10.3390/ijms20225533_

Round 1

Reviewer 1 Report

In their manuscript “Curcumin ameliorates benzo(a)pyrene-induced DNA damages in stomach tissues of Sprague-Dawley rats” Kyeong Seok Kim and co-authors test the alleviating effects of Curcumine on benzopyrene-induced tissue and DNA damage in several digestive tissues of rats. The authors find that co-administration of curcumine reduces the negative impact of  benzo(a)pyrene on liver, kidney and stomach, likely due to an inhibition of expression induction of two enzymes, CYP1A1 and CYP1B1, that stand at the beginning of a cascade leading to DNA damage. This effect is dose-dependent, although not all results fit neatly into the “the more the better” scheme.

Unfortunately in the manuscript version available for review part of the data is missing (tables 1, 2 and 3), so that it’s not possible to judge the manuscript completely. Thus, this review is taking only those results into account, that are mentioned in the text (without questioning these interpretations) or are shown in the figures.

The authors start their manuscript with an good introduction giving a brief overview over the connection of food-derived benzopyrene and cancer and also the current state of knowledge about protective effects of curcumine. The study design is clear and appropriate, although in retrospect it would have been good to include also only curcumin-treated animals into this study.

The result section begins with an abrupt jump into the detailed results. It would be helpful for the reader to get a brief summary/description of the conducted experiments, either at the beginning of the results section or the end of the intro. Currently the actual experiments are obvious only from the abstract or the methods section, that comes at the very end.

While the results are overall very consistent, their description in the text requires a bit of text editing:

line 76ff: “The treatment groups showed no significant difference in body and organs weights changes compared to those in the vehicle control group (Figure 1). However, liver weight was significantly reduced by co-administration of curcumin with BaP (Table 1)” These two sentences contradict each other, and fig. 1 gives only body weight, not organ weights. please define/explain all abbreviations, such as AST, ALT, BUN, CYP1A1 and CYP1B1 when using them first time

Also the discussion is well done, but contains small impreciseness in the text

line 201: “Although stomach and colon are known as non-target organs induced by BaP exposure” – not clear what this sentence means; stomach and colon cannot be induced, and if they are non-target organs, why would BaP cause any effect? line 212f: “may also be capable of inhibiting phase I enzymes.” – please explain what phase 1 enzymes are

The methods section describes the performed experiments clearly and is easy to follow. The comparison of the used BaP dose with a human example diet is very helpful and would absolutely deserve to be moved into the discussion (or intro or results) where it is more likely to be read.

The only improvement here would be the mentioning of catalogue numbers or otherwise more precise description of materials, especially for the ELISA kits: ”All other western blot  reagents were from Merk Millipore (Burlington, MA, USA).” and “All enzyme-linked immunosorbent assay (ELISA) kits were purchased from Cell Biolabs Inc.”

The figures are well done, but the figure legends require some additional information:

Fig.1: please indicate the statistical significance in some form. Is the difference between control and all treatments significant? The text states that there is no difference, but it would be good to give the appropriate analysis/p-values here in the figure legend.

Fig. 2: please explain how to interpret these pictures. State in which samples signs of pathophysiology are detected, and if possible, whether this is significant. Are these the only pictures taken, or are they some selected to be representative.

Fig.3 and the respective description in the text:

If BaP and metabolites were not eluted with the chosen conditions, why was this failed experiment included in the manuscript? It appears more sensible to change the experimental conditions, so that BaP and its metabolites are detectable. And if they were not detectable, how did the authors manage to analyze the levels of BaP, 3-hydroxy BaP (3-OH-BaP), 7- hydroxy BaP (7-OH-BaP) and BPDE, as given in Fig. 4?

Please include in Fig. 3 an explanation what the left four panels are.

Fig. 4: The statistics are unclear, as the legend mentions only #, but the figure contains *, ** and ##.  It would be good to also give the statistical difference between BaP only and BaP+Curcumin groups, as the potentially alleviating effect of Curcumin on BaP toxicity is the main focus of this study.

Fig. 5: This figure is clear, but it’s not obvious what is meant in the figure legend “Values are indicated by mean ± SD of 6 animals.”. Did this sentence end in here by accident or is there a graph missing quantifying the blot findings? What does the red arrow in the lower right panel indicate?

Fig.6 and 7: all fine

Author Response

Reviewer #1

Comment 1.Unfortunately in the manuscript version available for review part of the data is missing (tables 1, 2 and 3), so that it’s not possible to judge the manuscript completely. Thus, this review is taking only those results into account that are mentioned in the text (without questioning these interpretations) or are shown in the figures.

Response: We are sorry that this is our mistake, we added Table 1, 2, and 3 in the full text.

Comment 2.line 76: “The treatment groups showed no significant difference in body and organs weights changes compared to those in the vehicle control group (Figure 1). However, liver weight was significantly reduced by co-administration of curcumin with BaP (Table 1)” These two sentences contradict each other, and fig. 1 gives only body weight, not organ weights. Please define/explain all abbreviations, such as AST, ALT, BUN, CYP1A1 and CYP1B1 when using them first time

Response: Thank you for your valuable comments. We revised this part “The treatment groups showed no significant difference in body weights changes between BaP alone and in combination with BaP and curcumin groups. In addition, we corrected all abbreviations as your comments.

Comment 3. Also the discussion is well done, but contains small impreciseness in the text. Line 201: “Although stomach and colon are known as non-target organs induced by BaP exposure” – not clear what this sentence means; stomach and colon cannot be induced, and if they are non-target organs, why would BaP cause any effect?

Response: Thank you for your nice comment. Stomach is known as target organs induced by administration of benzo[a]pyrene. But ELISA results shows that higher fold increase in BPDE-DNA adduct formation compared to liver following oral administration of BaP. “Although there are linear correlation between mutation frequency and tumour incidence in target organs after exposure of animals to BaP, the levels of BaP-DNA adduct formation did not distinguish target (lung, spleen, and forestomach), from non-target organs (liver, colon and glandular stomach) in mice following oral administration of BaP [39]. This study was very similar to our present data that oral exposure to BaP significantly increased BPDE-I-DNA adduct formation in the stomach and liver tissues.”

Comment 4.212f: “may also be capable of inhibiting phase I enzymes.” – please explain what phase 1 enzymes are?

Response: Thank you for your nice comment. We corrected this part. “phase I enzyme-> BaP metabolism enzymes.”

Comment 5.The methods section describes the performed experiments clearly and is easy to follow. The comparison of the used BaP dose with a human example diet is very helpful and would absolutely deserve to be moved into the discussion (or intro or results) where it is more likely to be read. The only improvement here would be the mentioning of catalogue numbers or otherwise more precise description of materials, especially for the ELISA kits: ”All other western blot  reagents were from Merk Millipore (Burlington, MA, USA).” and “All enzyme-linked immunosorbent assay (ELISA) kits were purchased from Cell Biolabs Inc.”

Response: Thank you for your nice comment. We added catalog number of western blot reagents and ELISA kit (BPDE-DNA adduct and 8-OHdG kit).

Comment 6.The figures are well done, but the figure legends require some additional information:Fig.1: please indicate the statistical significance in some form. Is the difference between control and all treatments significant? The text states that there is no difference, but it would be good to give the appropriate analysis/p-values here in the figure legend.

Response: Thank you for your comment. According to your suggestion, we modified figure 1 statically.

Comment 7.Fig. 2: please explain how to interpret these pictures. State in which samples signs of pathophysiology are detected, and if possible, whether this is significant. Are these the only pictures taken, or are they some selected to be representative.

Response: Thank you for your comment. We already mentioned about pathological symptoms. “Arrows displayed cell infiltration, mononuclear cells, and multifocal cells. Arrowheads indicated focal nephropathy”

Comment 8.Fig.3 and the respective description in the text:If BaP and metabolites were not eluted with the chosen conditions, why was this failed experiment included in the manuscript? It appears more sensible to change the experimental conditions, so that BaP and its metabolites are detectable. And if they were not detectable, how did the authors manage to analyze the levels of BaP, 3-hydroxy BaP (3-OH-BaP), 7- hydroxyBaP (7-OH-BaP) and BPDE, as given in Fig. 4?

Response: Thank you for your comment. Sorry to inform you. This is our mistake. We revised it then “Chromatographic profiles were as almost baseline with elution of BaP and metabolites”

Comment 9.Please include in Fig. 3 an explanation what the left four panels are.

Response: Thank you for your comment. We added explanation.

Comment 10.Fig. 4: The statistics are unclear, as the legend mentions only #, but the figure contains *, ** and ##.  It would be good to also give the statistical difference between BaP only and BaP+Curcumin groups, as the potentially alleviating effect of Curcumin on BaP toxicity is the main focus of this study.

Response: Thank you for your comment. We revised figure 4 legend.

Comment 11.Fig. 5: This figure is clear, but it’s not obvious what is meant in the figure legend “Values are indicated by mean ± SD of 6 animals.”. Did this sentence end in here by accident or is there a graph missing quantifying the blot findings? What does the red arrow in the lower right panel indicate?

Response: Thank you for your comment. We revised figure 5 legend. “The western blot results represent three separate experiments.

Reviewer 2 Report

The manuscript by Kim and colleagues deals with the effect of curcumin on BaP-treated rats in the context of DNA-damage. The authors have focused on BaP-induced stomach cancer and the potential of curcumin to intervene in the cancer-inducing effect.

The manuscript is well written, however, there are some points that should be addressed by the authors:

All tables where the authors refer to in the text are missing (organ weights, serum levels of enzymes, focal nephropathy).

There are some earlier publications missing that investigated already an anti-carcinogenic effect of curcumin in vivo which should be included into the manuscript (e.g. Singh SV et al. Carcinogenesis 1998; Azmine et al. J Cancer Res Clin Oncol 1992).

It is not completely clear how the administration of the compound has been performed. Did the mice get additional oral treatments in case they received curcumin + BaP? If yes - have the control mice also received the second “mock” treatment. If not, this may explain the higher (although not significantly different) body weight in the control mice. At day 29 there is a drop in body weight in all groups unless the control group. Do the authors have any explanation? Have the oral gavage of the compound been performed every day for 30 days? Why have the authors chosen the indicated curcumin concentrations?

As BaP could be ingested via the food – why have the authors only focused on the stomach? As curcumin has a relatively low bioavailability one could postulate that high levels of curcumin reach the distal part of the gut. Maybe curcumin would also lower DNA-adduct formation in the intestine and potentially at even lower concentrations.

As curcumin has also been shown to increase phase II enzymes (which helps in excreting the more cancerous phase I enzyme products) and the transcription factor Nrf2 that is controlling their expression - Have the authors any information if Phase II enzymes were also induced in the BaP+Curcumin treated mice?

General: The resolution in Figure 2 should be increased.

Author Response

Reviewer #2

Comment 1.All tables where the authors refer to in the text are missing (organ weights, serum levels of enzymes, focal nephropathy).

Response: We added all Tables in the text.

Comment 2.There are some earlier publications missing that investigated already an anti-carcinogenic effect of curcumin in vivo which should be included into the manuscript (e.g. Singh SV et al. Carcinogenesis 1998; Azmine et al. J Cancer Res ClinOncol 1992).

It is not completely clear how the administration of the compound has been performed. Did the mice get additional oral treatments in case they received curcumin + BaP? If yes - have the control mice also received the second “mock” treatment. If not, this may explain the higher (although not significantly different) body weight in the control mice. At day 29 there is a drop in body weight in all groups unless the control group. Do the authors have any explanation? Have the oral gavage of the compound been performed every day for 30 days? Why have the authors chosen the indicated curcumin concentrations?

Response: Thank you for your comment. Benzo[a]pyrene and curcuim in corn oil was administered oral gavage. The treatment group only received curcumin (50, 100, and 200 mg/kg/day, up to total 30days).Therefore, we did not perform any other “mock”treatment group for our work. At that time (day 29), It appears to have been reduced by stress, not by the effect of the curcumin or Benzo[a]pyrene. Various concentrations of curcumin are used for oral administration (60 ~ 400 mg/kg/p.o). The doseof curcuminused in the rat model is 100 mg/kg/b.w. So we select the 3 doses (50, 100, and 200 mg/kg/b.w) for dose-dependently effect. (Ref, Protective Effects Of Curcumin Against Benzopyrene Induced Liver Toxicity In Albino Rats. Kolade and Oladiji., 2018; Combined effects of curcumin and piperine in ameliorating benzo(a)pyrene induced DNA damage. Sehgal et al., 2011; Study to Evaluate Molecular Mechanics behind Synergistic Chemo-Preventive Effects of Curcumin and Resveratrol during Lung Carcinogenesis.Malhotra et al., 2014; Curcumin modulates drug metabolizing enzymes in the female Swiss Webster mouse. Valentine et al., 2006)

Comment 3.  As BaP could be ingested via the food – why have the authors only focused on the stomach? As curcumin has a relatively low bioavailability one could postulate that high levels of curcumin reach the distal part of the gut. Maybe curcumin would also lower DNA-adduct formation in the intestine and potentially at even lower concentrations.

Response: Thank you for your nice comments. As you know, the role of smoked food in gastric carcinogenesis was suggested in the early 1960. Studies in Europe at the time showed that gastric cancer rates were highest in Finland and Iceland, where smoked fish and meat use was very high, which led to a further examination of smoked food and their polycyclic aromatic hydrocarbon (PAH) content in gastric carcinogenesis. Since then, benzo[a]pyrene and other PAHs formed in smoked food have been incriminated in many areas of the world with high gastric cancer rates. However, few studies have been performed with using the stomach as a target organ against from benzo[a]pyrene induced carcinogenesis.     

That’s right, as your comment, curcumin has a relatively low bioavailability from the stomach, However, we did not compared DNA-adduct formation in the intestine using low concentrations of curcumin.

Comment 4.  As curcumin has also been shown to increase phase II enzymes (which helps in excreting the more cancerous phase I enzyme products) and the transcription factor Nrf2 that is controlling their - Have the authors any information if Phase II enzymes were also induced in the BaP+Curcumin treated mice?

Response: Thank you for pointing in out. Phase II enzymes and Nrf2 are closely associated with the detoxification of BaP. As you know, the transcription factor Nrf2 antioxidant response pathway attenuates oxidative stress induced cell damage. And Phase II metabolic enzymes are playing an important role in detoxifies xenobiotics by increasing their hydrophilicity and enhancing their disposal. Phase II enzymes and Nrf2 have been studied for a long time for protective effect against mutagens and carcinogens by regulating through Keap1 (Kelch-like ECH related protein 1) / Nrf2 (nuclear factor erythrocyte 2 related factor 2) / ARE (antioxidative response factor). Furthermore, many researches show that curcumin has been effect to increase CYP1A1 activity and AhR–DNA binding and Nrf2 nuclear protein in response in cancer cells and mice, together with the induction of phase II enzymes [GST, NAD(P)H:quinone oxidoreductase-1 (NQO1), heme oxygenase 1 (HO-1)] activity. (Ref. Effect of curcumin on the aryl hydrocarbon receptor and cytochrome P450 1A1 in MCF-7 human breast carcinoma cells. Ciolino,H.P. et al. (1998); Curcumin activates the haem oxygenase-1 gene via regulation of Nrf2 and the antioxidant-responsive element. Balogun,E. et al; Involvement of Nrf2, p38, B-Raf, and nuclear factor-kappaB, but not phosphatidylinositol 3-kinase, in induction of hemeoxygenase-1 by dietary polyphenols. Andreadi,C.K. et al; Dietary curcumin modulates transcriptional regulators of phase I and phase II enzymes in benzo[a]pyrene-treated mice: mechanism of its anti-initiating action. Garg et al). Therefore, we did not focused phase II and Nrf2 pathway in this study.

Comment 5.General: The resolution in Figure 2 should be increased.

Response: Thank you for pointing in out. We revised figure 2.

Round 2

Reviewer 1 Report

Thanks to the authors for addressing all my concerns.

Here's only a a few small comments, mainly typong errors etc.:

abstract, line 26: BaP-diolepoxie. -> BaP-diolepoxide line 77ff: Animals treated with BaP or in combination with curcumin exhibited low body weight changes compared to the control. However, rats exposed to BaP or in combination with curcumin did not show significant changes in body weights (Figure 1). This is confusingly described. How about: “Animals treated with either BaP or BaP in combination with curcumin exhibited a slight, but significant reduction in body weight compared to controls. However, there was no difference among the treated groups.” line 128F: “The eluted as well as metabolite peak was confirmed based on the internal…” – here’s probably a BaP after “the eluted”missing Figure 5 legend: The western blot results represent three separate experiments. – it is unclear what three different experiments you mean – there’s two samples per condition on the shown blots, not three, so that cannot be meant. You also haven’t given any diagram or so, that would combine three different blots. Does this sentence potentially belong to figure 6? line 234: Although there are linear -> although there is

Author Response

Reviewer #1

Comment 1.abstract, line 26: BaP-diolepoxie. -> BaP-diolepoxide 

 Response: We are sorry that this is our mistake, we corrected.

Comment 2.line 77ff: Animals treated with BaP or in combination with curcumin exhibited low body weight changes compared to the control. However, rats exposed to BaP or in combination with curcumin did not show significant changes in body weights (Figure 1). This is confusingly described. How about: “Animals treated with either BaP or BaP in combination with curcumin exhibited a slight, but significant reduction in body weight compared to controls. However, there was no difference among the treated groups.”

Response: Thank you for your nice suggestion; we corrected this part according to your comment.

Comment 3.line 128F: “The eluted as well as metabolite peak was confirmed based on the internal…” – here’s probably a BaP after “the eluted”missing

Response: Thank you for your nice suggestion; we corrected this part according to your comment.

Comment 4. Figure 5 legend: The western blot results represent three separate experiments. – it is unclear what three different experiments you mean – there’s two samples per condition on the shown blots, not three, so that cannot be meant. You also haven’t given any diagram or so, that would combine three different blots. Does this sentence.

Response: Yes, we indicated the two samples per condition on the western blots. We revised this part “Representative bands of western blot for CYP1A1 and CYP1B1 were shown.”

Reviewer 2 Report

In the revised version of the manuscript, Kim and co-workers have addressed some of my comments. I still have some issues that should be clarified by the authors.

It is still not clear how the application procedure has been performed. Have the animals received curcumin via oral gavage for 30 days which was followed by only one oral application of BaP at the end? Or have the animals received both, 1st: curcumin by oral gavage and 2nd: BaP by oral gavage for 30 days? If the latter was the case, has the control group also received two oral treatments with vehicle after each other? If the control group received less gavages than the treatment groups it could be a lower level of stress for the control group being the reason for the observed differences e.g. in bodyweight. The authors refer to the fact that the body weight decreases at day 29 due to stress and not due to the treatment. In this case one has to ask what is the effect of the stress (that obviously was not present in the control group) and is this kind of stress possibly responsible for the observed results?

In Figure 5: The western blot potentially does not represent 3 individual rather than 1 representative western blot out of 3. This should be clarified.

There are earlier publications available regarding anti-carcinogenic activity of curcumin in vivo which have not been included into the present manuscript (e.g. Singh SV et al. Carcinogenesis 1998; Azmine et al. J Cancer Res ClinOncol 1992).

Author Response

Reviewer #2

Comment 1. It is still not clear how the application procedure has been performed. Have the animals received curcumin via oral gavage for 30 days which was followed by only one oral application of BaP at the end? Or have the animals received both, 1st: curcumin by oral gavage and 2nd: BaP by oral gavage for 30 days?

Response: Yes, BaP was administered orally to rats every day for 30 days after curcumin administration. Benzo[a]pyrene and curcuim in corn oil was administered oral gavage. The treatment group only received curcumin (50, 100, and 200 mg/kg/day, up to total 30days). We already mentioned detailed doses information of our animal excrement part. Page 9, line 284-287. “Animals in each group (n=6) were randomly divided into five groups: (1) Control group, whereby rats were treated with corn oil; (2) BaP-treated group, where rats were administered orally BaP (20 mg/kg) dissolved in corn oil; and (3) Co-treatment of BaP and curcumin groups, whereby curcumin (50, 100, or 200 mg/kg) was administered orally before BaP (20 mg/kg) exposure for 30 days.”

Comment 2. If the latter was the case, has the control group also received two oral treatments with vehicle after each other? If the control group received less gavages than the treatment groups it could be a lower level of stress for the control group being the reason for the observed differences e.g. in bodyweight. The authors refer to the fact that the body weight decreases at day 29 due to stress and not due to the treatment. In this case one has to ask what is the effect of the stress (that obviously was not present in the control group) and is this kind of stress possibly responsible for the observed results?

Response: Thank you for your kind comments. In the present study, both control group BaP alone group and BaP + curcumin groups were orally administered twice daily. Therefore, stress effects were not present in the control group as compared with others groups because BaP alone treatment groups also administered orally two times per every days.

Comment 3. In Figure 5: The western blot potentially does not represent 3 individual rather than 1 representative western blots out of 3. This should be clarified.

Response: Yes, we indicated the two samples per condition on the western blots. We revised this part “Representative bands of western blot for CYP1A1 and CYP1B1 were shown.”

Comment 4. There are earlier publications available regarding anti-carcinogenic activity of curcumin in vivo which have not been included into the present manuscript (e.g. Singh SV et al. Carcinogenesis 1998; Azmine et al. J Cancer Res Clin Oncol 1992).

Response: Thank you for your kind suggestion, we added these references (27,28).